# Cell Phone Social Media Use and Psychological Well-Being in Young Adults: Implications for Internet-Related Disorders

**DOI:** 10.3390/ijerph20021197

**Published:** 2023-01-10

**Authors:** Suresh C. Joshi, Steven Woltering, Jay Woodward

**Affiliations:** 1Department of Digital Learning, SGT University, Budhera, Gurugram-Badli Road, Gurugram 122505, Haryana, India; 2Educational Psychology, Texas A&M University, College Station, TX 77843, USA

**Keywords:** cell phone social media use, internet-related disorders, behavioral addiction, young adults, psychological well-being

## Abstract

Internet-related disorders are on the rise and increasing cell phone social media use may be one of the reasons for these disorders. To better understand internet-related disorders, we aim to explore the psychological and social aspects of cell phone social media behaviors. We hypothesized that, according to humanistic theories of positive functioning, cell phone social media connectedness to self (engagement, interest, pleasure, sense of enjoyment, meaningfulness, purposefulness, optimism, acceptance, and feeling accomplished) would relate positively to psychological well-being of undergraduate students. We also hypothesized that, according to Maslow’s hierarchy of needs, cell phone social media connectedness to others (affiliation, feeling rewarded, being liked by others, and contributions to the well-being of others) would relate positively to the psychological well-being of undergraduate students. During the fall of 2019, 523 (75.4% female) undergraduate students from a large public university participated in this study by completing validated quantitative surveys on their cell phone use and psychological well-being. Spearman’s rho and ordinal logistics were implemented to analyze the findings. Correlational data showed that cell phone social media connectedness to self and cell phone social media connectedness to others were positively associated with the psychological well-being of undergraduate students. Ordinal logistics showed higher odds of psychological well-being occurring with cell phone social media connectedness to self and cell phone social media connectedness to others. Cell phone social media connectedness to self significantly predicted psychological well-being with the medium effect, whereas cell phone social media connectedness to others was not a significant predictor of the psychological well-being of undergraduate students. An increase in cell phone social media connectedness to self and an increase in cell phone social media connectedness to others of undergraduate students helped them improve their psychological well-being. Cell phone social media connectedness to self significantly predicted but cell phone social media connectedness to others did not predict the psychological well-being of undergraduate students, which may have implications for the research pertaining to behavioral addiction and may help better understand internet-related disorders.

## 1. Cell Phone Social Media Use (SMU) in Young Adults

Internet users spent around 145 min per day on social media [1], and 80% of them accessed it through cell phones [2]. Young adults (YA’s) aged 18–29 with an average social media usage of 181 min per day, were the largest demographic of users [3,4,5]. Social media apps on cell phones were used for various purposes including access to information, ease of communication, and freedom of expression, but mostly for social networking [6]. It was found that cell phone social media help YA’s feel engaged, connected, and interested [7]. It was also found that cell phone social media use (SMU) influences the social life of YA’s [1] as they express their feelings, positive or negative, about their relationships on social media platforms [8,9]. There may be the case that cell phone SMU influences the psychological well-being (PWB) of YAs, which may further be linked to internet-related disorders. The present study, therefore, aims to investigate cell phone SMU and PWB in a population of YAs.

## 2. Existing Literature on Cell Phone SMU and PWB

Previous studies have provided multiple arguments for the impact—positive or negative—of SMU on the PWB of YAs.

Concerning positive impact, the use of social media for communication and connectedness has been associated with psychological benefits in YAs [6]. Cell phone SMU for communication and self-disclosure was positively related to the PWB of YAs [9]. Smith et al. [10] found that communicating through social media was a “legitimate means of developing social connections” (p. 12) and was a key factor in fostering a sense of belonging among YAs. Social isolation was detrimental to several mental health problems, including suicidal ideation, but SMU helped reduce social isolation thus improving the levels of PWB in YAs [11]. YAs were “happier when on their devices, particularly when they were alone but preferred to be with people” [12]. Following friends on social media was helpful in coping with loneliness, however, following strangers was not [13]. SMU helped YAs foster social connectedness and reduce FOMO (fear of missing out) resulting in improved levels of PWB [13,14]. SMU also helped YAs improve their social relationships through caring for others [15]. The socio-psychological prosperity of more extroverted SMUsers was found to be high as they had increased psychological benefits compared to that of less extroverted users [15,16].

Concerning negative impact, cell phone SMU in YAs has been associated with “idleness and perceived timelessness while staying physically immobile” [17] (p. 10). SMU was also found to be problematic for life satisfaction as studies indicated it increased loneliness in YAs [18]. The increased loneliness further escalated cell phone addiction in YAs [19]. Using specific social media platforms such as Facebook and Instagram was correlated with repetitive negative thinking and lower self-esteem, which resulted in decreased well-being in the young adult population [20]. When addiction to Instagram was severe, it had a significant negative impact on the loneliness and shyness of YAs [21]. Problematic CPU and SMU were associated with anxiety and depression in a sample of college students comprised of YAs [22]. Female YAs “whose self-worth was dependent on social media feedback reported lower levels of resilience and self-kindness and higher levels of stress and depressive symptoms,” and therefore were more prone to reduced PWB [23]. Compulsive SMU was associated with high FOMO and low PWB in YAs, with a stronger association with working professionals than college students [24]. In sum, cell phone SMU has been associated with the positive and negative aspects of PWB in YAs, however, the clarity of the link between the two variables is completely missing.

Noteworthy is the fact that, in previous studies, the PWB was measured through the associated variables but was not measured as an independent variable. For example, the paragraphs (above) describing the positive and negative impact of SMU on the PWB comprise studies examining variables such as psychological benefits, sense of belonging, social isolation, social relationships, idleness, perceived timelessness, life satisfaction, and self-esteem. As per the humanistic theories of affective functioning [25], the PWB is different from subjective well-being, therefore, should be measured separately. Measuring PWB using indirect variables may not provide the correct assessment of it [26]. In fact, convinced of the need for PWB measures, Diener et al. [27,28] have developed a PWB scale, named the flourishing scale (FS), which was based on the psychological theories of human flourishing. The FS was meant for measuring all the aspects of positive human functioning and social-psychological prosperity such as meaning and purpose, supportive and rewarding relationships, engagement and interest, contribution to the well-being of others, competency, self-acceptance, optimism, and being respected [27,28]. Two things were pertinent for the correct measurement of cell phone SMU and PWB (1) the operationalization of cell phone SMU in alignment with the aspects of positive human functioning and social-psychological prosperity, and (2) the use of the FS for the measurement of PWB. The present study attempts to fulfill both requirements. For that reason, in the following sections, we present the operationalization of cell phone SMU in alignment with the underlying theories.

## 3. The Operationalization of Cell Phone SMU Variables

Based on humanistic theories, we operationalized cell phone SMU in terms of two variables (i) how users feel—connectedness to self, and (ii) how they express their feelings—connectedness to others. The first variable was described as how cell phone SMU makes users feel when they see a post on social media (Instagram, Twitter, Facebook, Snapchat, LinkedIn, etc.). This variable was aligned with the dimensions of effective human functioning [27] and termed cell phone social media connectedness to self (CPSM-S).

The CPSM-S measures users’ connectedness to self along the dimensions of effective human functioning: engagement, interest, pleasure, sense of enjoyment, meaningfulness, purposefulness, optimism, acceptance, and feeling accomplished [27]. The second variable was described as how cell phone social media users express their feelings about others’ posts or the way they perceive others’ responses to their posts on social media, i.e., likes, shares, and comments followed by emojis, GIFs (graphics interchange format images), or stickers. This variable was aligned with the dimensions of positive social relationships [29] and termed cell phone social media connectedness to others (CPSM-O). The CPSM-O measures users’ perceptions along the dimensions of positive social relationships: affiliation, feeling rewarded, being liked by others, and contributions to the well-being of others [29]. Both effective human functioning and positive social relationships were the key indicators of PWB [27].

## 4. Underlying Theories

Two theories were attributed to the association between cell phone SMU variables (i.e., CPSM-S and CPSM-O) and PWB. The first theory, the six-factor model of psychological well-being [25], was attributed to the association between cell phone SMU and CPSM-S. As per the theory, six theory-guided dimensions led to effective human functioning. These dimensions were: self-acceptance, positive relations with others, autonomy, environmental mastery, purpose in life, and personal growth constitute the positive psychological functioning of humans. Optimism was included as an additional dimension of effective human functioning [30]. The second theory, the Maslow hierarchy of psychological needs [29], was attributed to the association between cell phone SMU and CPSM-O. As per the theory, psychological needs included esteem needs, belongingness, and love needs, which led to the dimensions of positive social relationships. These dimensions were: connectedness, social interaction, affiliation, being liked by others and giving and receiving [31]. In sum, the dimensions of effective human functioning and positive social relationships were aligned with CPSM-S and CPSM-O, thus may affect the PWB.

## 5. Study Rationale, Problem Statement, and Research Hypotheses

Previous studies have examined SMU and PWB, however, the correlation between these two variables is unclear. SMU scale comprising up-to-date quantifiable measures was not used. Moreover, PWB was measured in terms of associated variables, and a scale particularly dedicated to assessing PWB was not used. In addition, no theoretical support was provided for the correlation between SMU and PWB. The present study was, therefore, proposed to examine, in a theory-driven manner, the correlation between cell phone SMU and PWB of YAs using up-to-date quantifiable measures.

The research hypotheses of the study were as follows:

**Hypothesis 1 (H1).** *We expect, according to humanistic theories of positive functioning, cell phone social media connectedness to self (CPSM-S) to relate positively to the PWB of undergraduate students*.

**Hypothesis 2 (H2).** *We expect, according to Maslow’s hierarchy of needs (psychological needs, i.e., belonging and esteem needs), cell phone social media connectedness to others (CPSM-O) to relate positively to the PWB of undergraduate students*.

## 6. Materials and Methods

### 6.1. Participants

Five hundred twenty-three undergraduate students (75% female), between 18 and 29 years old with an average age of 20.01 years (SD = 3.18), participated in this study. The participation rate was 1.1% of the total headcount at the university at the time of the survey. This sample was reflective of the university population at a 95% confidence level with a margin of error of ±4.25%. This means if the CPU study survey had been completed by the entire university population, 95% of the time, 45,836 (95.75%) of the undergraduate students would have picked the same answers that were picked by the CPU study sample. In other words, if the CPU study survey is repeated using the same methods, 95% of the time the CPU study sample statistics will represent the current university population parameters with a ±4.25% margin of error. In survey-based studies, with random sampling, a margin of error of up to ±8% (95% confidence level) is acceptable [32].

The sample of this study was ethnically diverse, which comprised Caucasian (49%) Latinx (24%), Asian (19%), African American (3%), and Native American (1%) student population. In the survey, 3% of the undergraduate students were identified as “other” and the remaining 1% preferred not to answer. The sample of this study was also diverse in terms of the years they have been in college, i.e., 38% of undergraduate students were incoming freshmen, 19% were sophomores, 17% were juniors, 14% were seniors, and 13% were returning seniors. These undergraduate students were from 14 different majors, including engineering (29%), education and human development (9%), veterinary medicine and biomedical sciences (7%), science (9%), liberal arts (16%), business (7%), and agriculture and life sciences (17%).

### 6.2. Procedures

An online quantitative survey, comprising both CPU and PWB scales along with the demographic questions, was shared with the participants through the university’s listserv. The survey was developed considering best practices for constructing online assessment tools [33,34]. Informed consent was obtained from the participants before going to the actual survey. Participants had a chance to read all the necessary information before signing the informed consent. Thereafter, the participants were provided the access to the questionnaire, which was compatible with both cell phones and other electronic devices such as laptops. Participation in this study was completely voluntary and participants had the option to quit the survey anytime they wanted.

### 6.3. Measures

#### 6.3.1. PWB Questionnaire (Flourishing Scale)

A validated 8-item flourishing scale (FS) (Cronbach’s alpha = 0.87), developed by Diener et al. [28], was used to measure PWB. The FS measured the core attributes of positive human psychological functioning, i.e., purpose and meaning, supportive relationships, engagement, competence, optimism, self-acceptance, being respected, and contribution to the well-being of others. The questionnaire used a five-point Likert scale ranging from “1 = Strongly disagree” to “5 = Strongly agree”, with total scores ranging from 8 to 40. A higher score indicated better psychological resources.

As per Diener et al. [28], FS has good psychometric properties as it was highly similar to other psychological well-being scores although it does not assess individual components of social PWB. All items of the FS had strong internal consistency in several other previous studies (Cronbach alpha’s of 0.80 and 0.87) [16,35]. The present study (*n* = 523) also tested the reliability of the FS and found items to be reliable (Cronbach’s alpha = 0.89).

#### 6.3.2. CPU Questionnaire

A 12-item comprehensive cell phone SMU scale was used with the up-to-date questions, which were apt for research hypotheses/questions. In this scale, eight items measured CPSM-S on a scale ranging from “1 = Never” to “4 = Always”, with a total score ranging from 8 to 32. The sample items were: (i) Social media apps make me feel engaged and connected to my peers, (ii) I find social media apps interesting, and it satisfies a curiosity. Four items measured CPSM-O on a scale ranging from “1 = Never” to “4 = Always”, with a total score ranging from 4 to 16. The sample items were: (i) feel like I contribute to the well-being of others when I actively respond to OTHERS’ posts on social media, (ii) feel liked when others respond to YOUR posts on social media (refer to Appendix B for the complete survey). This survey was completely based on the scales used and validated in previous studies. While updating the questions, the psychometric principles, and the best practices for constructing an online assessment tool [33] were strictly followed. Most of the items in the scale were adapted from previous studies, however, were modified linguistically to make them clearer and more understandable.

The translational validity (i.e., content validity and face validity) of the instrument was assessed before using the survey [36,37]. Translation validity determines the degree to which constructs were accurately operationalized, using face and content validity [36]. Face validity was assessed and updated based on feedback from two specialists working at the University Center for the Advancement of Literacy and Learning. The content validity was assessed by faculty experts from the departments of Communication and English, and the instrument was found to be valid. The criteria for content validity included feedback regarding whether items effectively captured what was intended to measure, consistency of content and language use for modified/extended items, as well as alignment of items within constructs.

The instrument’s reliability was tested using the final sample (*n* = 523). The overall scale exhibited strong reliability (Cronbach’s alpha = 0.92). The subscales, CPSM-S and CPSM-O had excellent (Cronbach’s alpha = 0.90) to good (Cronbach’s alpha = 0.87) internal consistency, respectively. Please refer to Appendix A and Appendix B for more details on the internal consistency of the items as well as information on any modification on sources that were used.

### 6.4. Data Analysis

We have used the statistical package SPSS for Windows (Version 25.0, Chicago, IL, USA) for all analyses. Concerning the nature of the variables, both the independent variables, i.e., CPSM-S and CPSM-O, were continuous with ordinal ranked data, which was non-normal and skewed. The dependent variable, i.e., PWB, was also continuous with ordinal, non-normal, skewed, and heteroscedastic data. Non-parametric inferential statistics, i.e., Spearman rank-order correlation were used to analyze the correlation between cell phone SMU variables (CPSM-S and CPSM-O) and PWB. Parametric inferential statistics, i.e., ordinal logistic regression was used to examine the relative contribution of other variables to the PWB of undergraduate students. The partial eta squared was used to determine the effect size between the groups.

A control analysis was conducted for predicting variables CPSM-S and CPSM-O, which included the ceiling effect, floor effect, and test of multicollinearity. No ceiling or floor effect was found for CPSM-S (for the lowest score—3.4%, for the highest score—1.1%) and CPSM-O (for the lowest score—6.1%, for the highest score—6.1%) as the frequency percentage of respondents achieving the highest or lowest possible score was less than 15% for both the variables. None of the predictors in this study were correlated with other variables as the variance inflation factor (VIF) for both predictors was less than 3 (VIF_CPSM-S = 1.85; VIF_CPSM-O = 1.83).

The test of normality, the test of homoscedasticity, and the test of proportional odds were conducted for the outcome variable (PWB) as a part of the control analysis. The data was found to be non-normal (Kolmogorov-Smirnov test statistics = 0.062, df = 523, *p* < 0.001; Shapiro-Wilk test statics = 0.976, df = 523, *p* < 0.001), skewed (skewness = −0.83), and heteroscedastic (cone shaped residual plot). For this data, the test of proportional odds was found to be significant (Chi-Square = 35.2185, df = 523, *p* < 0.01) indicating that proportional odds do not hold. 

The difference between the groups was tested using one-way ANOVA and post-hoc analyses at the significance level of 0.01. Effect sizes between the groups were computed using partial eta squared [35,36].

## 7. Results

### 7.1. Descriptive Statistics

With a mean score of 2.09, “occasional” CPSM-S encounters were reported by undergraduate students on a scale ranging from 1 to 4, with 1 being “Never” and 4 being “Always” (Table 1). Undergraduate students also reported encounters with CPSM-O on a scale ranging from 1 to 4, with 1 being “Never” and 4 being “Always”, but with a measure varied between “occasional” and “often” (score; 2.24–2.54) (Table 1).

### 7.2. Inferential Statistics

Inferential statistics will be presented in three sections. The first section will describe the group differences. The second and third sections will describe the results pertaining to the first and second hypotheses.

There was a significant difference for CPSM-S between the group means of ethnicity (*F* (6, 518) = 4.935, *p* < 0.01, eta squared = 0.05), as determined by a one-way ANOVA. Caucasian undergraduate students (2.36 ± 0.76) had lower CPSM-S compared to Asian undergraduate students (1.99 ± 0.59). Ethnicity had a statistically significant (*p* < 0.01) effect, with small and medium effect sizes, on CPSM-S activities of undergraduate students. Asian undergraduate students had higher CPSM-S scores for all activities mentioned compared to Caucasian undergraduate students. Several independent variables such as sex, ethnicity, year in college, and college had no statistically significant (*p* < 0.01) effect on the CPSM-O.

Analysis indicated that the first hypothesis H1 was supported. The CPSM-S was positively correlated with the PWB of undergraduate students (Spearman’s coefficient = 0.172, *p* < 0.001) (Table 2). The crude odds ratio indicated that there are higher odds of PWB happening with CPSM-S of undergraduate students (Exp (B) = 1.798, *p* < 0.001) (Table 3). Further, the correlation between CPSM-S and PWB was confirmed by the likelihood chi-square ratio (chi square (1) = 15.129, *p* < 0.001). The relationship holds well within the model as the CPSM-S parameters fit well in the proportional odds ratio independence assumption (as *p* = 0.985). Chi-square was used for testing the fitting of the model, and it was found that the model significantly fits the null model (Omnibus test chi-square (12) = 53.291, *p* < 0.001). Controlling other variables had a positive impact on the outcomes, which improved the ability of the model to predict the correlation. CPSM-S strongly predicted PWB (Exp (B) = 1.913, *p* < 0.001) and may have a medium effect on undergraduate PWB (*F* (24, 495) = 2.054, *p* < 0.01, partial eta squared = 0.09).

Analysis also indicated that the second hypothesis H2 was supported. The CPSM-O was positively correlated with the PWB of undergraduate students (Spearman’s coefficient = 0.126, *p* < 0.001) (Table 2). The crude odds ratio indicated that there are higher odds of PWB occurring with the CPSM-O of undergraduate students (Exp (B) = 1.352, *p* < 0.01) (Table 3). However, the relationships did not hold within the model (as *p* = 0.000) as the CPSM-O parameters did not fit well in the proportional odds ratio independence assumption. The likelihood chi-square ratio (chi square (1) = 0.069, *p* = 0.792) and adjusted odds ratio were not statistically significant (Exp (B) = 1.036, *p* = 0.792) for the model, and when factored in the controlling variables, the correlation was not as strong to predict the PWB. The CPSM-O was not a significant predictor of the PWB, however, and had a medium effect on the PWB of undergraduate students (*F* (12, 507) = 1.887, *p* < 0.01, partial eta squared = 0.04).

## 8. Discussion

Results concerning the first hypothesis indicated that an increase in CPSM-S such as engagement, interest, pleasure, sense of enjoyment, meaningfulness, purposefulness, optimism, acceptance, and feeling accomplished increased the PWB of undergraduate students. The outcomes of this hypothesis were crucial as this study was the first one, as per our knowledge, that assessed CPSM-S using measures aligned to the states of effective human functioning. The descriptive statistics of CPSM-S (Table 1) indicated that the occasional use of cell phones for social media (Instagram, Twitter, Facebook, Snapchat, LinkedIn, etc.) helped undergraduate students feel connected, competent, and optimistic. Undergraduate students found social media apps interesting and meaningful, and the use of social media on cell phones often helped them feel pleasure and a sense of enjoyment, a sense of purpose and fulfillment, as well as a sense of belonging and acceptance, with Asian undergraduate students, as compared to Caucasian undergraduate students, having higher CPSM-S scores. These results align with the previous research [15], which found “connecting with others” as one of the motives for CPU. Moreover, this study supported the notion that CPU helps reduce loneliness [15], as the use of cell phones for social media helped undergraduate students feel engaged and connected.

The results from the ordinal logistic indicated that the increased levels of CPSM-S helped undergraduate students rank themselves as having more psychological resources and strengths. These outcomes endorsed previous research [9] that found communicative CPU, including social media, beneficial for PWB. Further, considering the fact that the use of cell phones for emotion regulation affects well-being positively [6], and the locus of control impacts satisfaction with life positively [38], it can be concluded that the use of cell phones helps undergraduate students improve their socio-psychological prosperity. Referring to the descriptive analyses of this study (Table 1), CPSM-S helped undergraduate students feel competent and accomplished, and moreover, had a sense of purpose and fulfillment. Altogether these results show that the CPSM-S hypothesis supported the Six-factor model of psychological well-being [25], and it can be established that the use of cell phones for social media helps undergraduate students improve their PWB.

Results concerning the second hypothesis indicated that an increase in CPSM-O, such as affiliation, feeling rewarded, being liked by others, and contributions to the well-being of others (by liking, sharing, loving, using emojis, posting GIFs, attaching stickers, etc.), feeling liked, and feeling rewarded, increased the PWB of undergraduate students. The descriptive statistics of CPSM-O (Table 1), which indicated that undergraduate students feel liked when others respond to their posts on social media more frequently, supported the effect size analysis of CPSM-O. The descriptive analyses also suggested that cell phone social media engagements helped undergraduate students feel connected, rewarded, and contribute to the well-being of others when they actively respond to others’ posts on social media more than occasionally. It may be due to these reasons that the socio-psychological prosperity of more extroverted users was higher than that of less extroverted users [15]. For these reasons, it can be concluded that the correlation between CPSM-O and PWB supports Maslow’s hierarchy of psychological needs [29], which includes the feeling of belongingness including affiliation, social interaction, friendship, giving and receiving, and contributions to the well-being of others.

Ordinal logistic regression indicated that increased levels of CPSM-O did not predict whether or not undergraduate students ranked themselves as having more psychological resources and strengths. These outcomes resonate with the outcomes of the previous research [16,38,39] on cell phone addiction (smartphone addiction in the case of smartphone users) and PWB. Kumcagiz and Gunduz [16] have reported that high smartphone users had lower levels of PWB than low smartphone users. It might have been the case that overwhelming CPSM-O jeopardized their social relationships, as high CPU and excessive online communication inversely affected well-being and PWB, respectively [40,41]. Limited research in this area restricts justification and caution while interpreting results. Future research with more quantifiable measures of CPSM-O would help us understand whether CPSM-O predicts the PWB of undergraduate students.

In previous studies, the cell phone SMU of young adults was examined with health variables along with subjective well-being; however, a direct association between cell phone SMU and PWB was left unexplored. The subjective well-being was further assessed in terms of satisfaction with life [15,42,43], emotional and relational well-being [44], and overall well-being [6,42], and was found to be correlated with cell phone SMU. Previous studies have also claimed to investigate cell phone SMU and PWB; however, they either missed assessing a direct correlation between CPU and PWB or ended up with conflicting outcomes. For example, Chan [40] investigated the use of cell phones in terms of four CPU dimensions (voice communication, online communication, information-seeking activities, and time-pass activities), and focused only on the emotional aspect of well-being. Chen and Li [9] examined how communicative uses of cell phones, including friending self-disclosure, may help predict PWB through bonding and bridging social capital. Further, Murdock, Gorman, and Robbins [45] investigated how co-rumination via cell phones moderates the association between perceived interpersonal stress and PWB. Examples of conflicting outcomes came from the following two studies (1) the first study [15] showed a negative correlation between CPU and PWB variables, such as loneliness and depression, (2) the second study [16] showed a positive correlation between low CPU and the improved levels of PWB. Similarly, in the first study, the socio-psychological prosperity of more extroverted cell phone users was found to be higher than that of less extroverted cell phone users, and in the second study, high CPU was found to be related to low levels of PWB. The present study filled the gap by investigating the cell phone SMU and PWB of young adults using up-to-date quantifiable measures. Moreover, the present study resolved the existing conflict by investigating a direct relationship between cell phone SMU and PWB.

One reason why previous studies [9,15,16,40,41,42,43,44,45] lacked theoretical support perhaps that a correlation between CPU and PWB was not investigated in these studies. The present study examined two cell phone SMU hypotheses: CPSM-S, and CPSM-O, and both hypotheses were supported by the existing developmental theories. The CPSM-S hypothesis was supported by the humanistic theories of positive functioning, and the CPSM-O hypothesis was supported by two theories: Maslow’s hierarchy of needs and self-determinant theory. These hypotheses were attributed to the correlation between cell phone SMU and PWB of the young adult demographic, which has added new knowledge to the literature in developmental sciences research.

Combining everything together, cell phone social media use may not contribute to internet-related disorders as both variables (CPSM-S and CPSM-O) were correlated positively to the PWB of YAs. However, the catalytic role of CPSM-O in behavioral addiction and internet-related disorders cannot be ruled out as CPSM-O did not predict the PWB of YAs. It may be the case that YAs spend more time to fulfill CPSM-O, as a result, indulge in excessive online communication, which makes them the victim of behavioral addiction and internet-related disorders. More research is needed in this area, especially to explore CPSM-O. A study examining the inter-variable interactions between CPSM-O, behavioral addiction, and internet-related disorders is warranted.

## 9. Conclusions

The purpose of this study was to investigate, in a theory-driven manner, using up-to-date quantifiable measures, the correlation between cell phone SMU and PWB of YAs. Two hypotheses were developed. The first hypothesis was that according to humanistic theories of positive functioning, CPSM-S relates positively to the PWB of undergraduate students. The second hypothesis was that according to Maslow’s hierarchy of needs (psychological needs, i.e., belonging and esteem needs), CPSM-O relates positively to the PWB of undergraduate students. The analyses of the study concluded that CPMS-S significantly predicted psychological well-being with the medium effect, whereas CPSM-O was not a significant predictor of the psychological well-being of undergraduate students. An increase in CPSM-S and an increase in CPSM-O of undergraduate students helped them improve their psychological well-being. Moreover, CPSM-S significantly predicted but CPSM-O did not predict the psychological well-being of undergraduate students, which may have implications for the research on behavioral addiction and may help better understand internet-related disorders. 

YAs might have been spending too much time on cell phone social media during class or study-related activities, which might have increased the total time spent on cell phones during academic activities. Similarly, YAs might have been spending too much time on cell phone social media just before sleep, which might have increased their sleep latency and sleep difficulty thus worsening the overall sleep quality. Previous studies indicated that switching between cell phones and academic tasks reduces their academic performance [46]. Also, the use of cell phones just before going to sleep reduces their sleep quality thereby impacting their PWB [47,48]. In a way, excessive cell phone SMU might contribute to internet-related disorders, and the related studies might help understand the underlying mechanism in behavior addiction concerning CPU.

## 10. Limitations

The results of this study should be interpreted keeping its limitations in mind. Firstly, the outcomes should be restricted to correlational studies and should not be extended to the studies concerning causality. Secondly, the study sample may be the reflection of some socioeconomic and cultural specificities of university students from the Southwestern region as the sample comprised undergraduate students from a single public university in the Southwestern United States. Thirdly, the results may be applicable to college students and may have limited applicability for non-college CPUsers. Fourthly, the outcomes may have more applicability for female college students as the sample suffered from overrepresentation for female participants (75%). Lastly, recall bias [49] due to self-reported measures may be one of the key limitations of this study.

## 11. The Significance of the Study and the Practical Applications of the Outcomes: Implications for Future Research

This study is of high significance for the researchers and policymakers as the study (1) operationalized cell phone SMU variables (i.e., CPSM-S and CPSM-O), (2) used up-to-date quantifiable cell phone SMU measures, (3) explored cell phone SMU and PWB of the young adult from the current population, (4) examined the direct relationship between the cell phone SMU variables and PWB of young adults, and (5) provided theoretical support to the association between cell phone SMU variables (i.e., CPSM-S and CPSM-O) and PWB. In addition, this study has direct practical applications for cell phone SMU users. Social media is the biggest virtual platform where young adults portray their lives to the public, and cell phones are the most accessible devices suitable for that purpose. The outcomes from the present study will serve as a guiding document for the researchers working on cell phone SMU. The data on CPSM-S will inform them about how a cell phone makes participants feel from a social media standpoint (Instagram, Twitter, Facebook, Snapchat, LinkedIn, etc.) along different dimensions: engagement and connectedness, interest, pleasure, and sense of enjoyment, meaningfulness, purposefulness, optimism, belongingness, and acceptance, and competence and feeling accomplished. The data on CPSM-O will tell them how young adults perceive a response to their posts and their responses to others’ posts on cell phone social media. Moreover, CPSM-O data will educate about the feelings of connectedness, being liked by others, reward, and contributing to the well-being of others based on responses with social media apps (Instagram, Twitter, Facebook, Snapchat, LinkedIn, etc.). In a nutshell, the outcomes informed researchers on how cell phone SMU shall be used as a potential tool for fulfilling psychological needs.

## Figures and Tables

**Table 1 ijerph-20-01197-t001:** The descriptive statistics of continuous variables age, CPSM-S, CPSM-O, and PWB (*n* = 523).

	Minimum	Maximum	Mean ± SD	Mode	Median (IQR)
Age	18	29	20.19 ± 3.18	18.00	
CPSM-S	1.00	4.00	2.09 ± 0.65	2.00	2.00 (0.86)
CPSM-O	1.00	4.00	2.42 ± 0.80	2.00	2.50 (1.00)
PWB	8	40	31.62 ± 5.54	32.00	

*Note.* CPSM-S = The use of cell phones for social media feeling, CPSM-O = The use of cell phones for social media response, PWB = psychological well-being.

**Table 2 ijerph-20-01197-t002:** Nonparametric correlations (*n* = 523).

		CPSM-S	CPSM-O	PWB
Spearman’s rho	CPSM-S	1.00		
	CPSM-O	0.657 **	1.000	
	PWB	0.172 **	0.126 **	1.000

*Note.* CPSM-S = cell phone social media feeling, CPSM-O = cell phone social media response, PWB = psychological well-being. ** *p* < 0.01 (2–tailed).

**Table 3 ijerph-20-01197-t003:** Ordinal logistic regression analyses showing the relationship between cell phone social media use variables (CPSM-S and CPSM-O) and psychological well being (PWB) (*n* = 523).

Dependent Variable	Independent Variable	Odds Ratio	
		Crude OR (95% CI)	Adjusted OR (95% CI) ^a^
PWB			
	CPSM-S	1.798 ** (1.413–2.286)	1.913 ** (1.379–2.654)
	CPSM-O	1.352 * (1.115–1.641)	1.036 (0.799–1.343)

*Note.* CPSM-S = cell phone social media feeling, CPSM-O = cell phone social media response. ^a^ adjusted for sex, ethnicity, colleges, age, years in college, CPSM-S, and CPSM-O. * *p* < 0.01; ** *p* < 0.001.

## Data Availability

Not applicable.

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
