# Peer review of "Cell Phone Social Media Use and Psychological Well-Being in Young Adults: Implications for Internet-Related Disorders"

_ijerph, 2023, doi:10.3390/ijerph20021197_

Round 1

Reviewer 1 Report

This study is meaningful, but there were problems with data analyze and have some minor mistakes.

1.     In Section 4, the two theories both direct to CPSM-O, which was confusing.

2.     In Section6.1, the mean of participants’ age should have the same decimal with the standard deviation.

3.     In Section6.3, please add the scorning method of the three scales.

4.     In Section6.3.2, Please give at least one example of CPMU-S and CPMU-O respectively.

5.     In Section 7.2, the mean and standard deviation were calculated of PWB, CPMU-S and CPMU-O, which mean the three variables are continuous variables. I was wonder why using Spearman Rho and Ordinal logistic regression to analyze the relationship between the variables? In the sentence “The relationship holds well across all the PWB categories within the model (Line 258)”, What’s the categories of PWB? Why PSW have categories and how classify?

6.     In Line 262, please point out the control variables used in the analysis.

7.     In appendix B, as the items of two scales were adapted from other scales, please clearly presented the origin of each specific items; the “Reliability and Validity Evidence” part gave the reliability of origin scales could not prove for the reliability of scales developed in the study.

Author Response

Please refer to the rebuttal letter attached herewith.

Reviewer 2 Report

The article entitled “Cell Phone Social Media Use and Psychological Well-Being in Young Adults: Implications for internet-related disorders” is an engaging and timely contribution to the literature on the relationship between technical devices and well-being. However, a few changes need to be made before publication is endorsed.

The abstract needs to be simplified to ensure interest from a broad readership. My modest advice is to describe the results in plain English without relying on the numeric values of the results obtained or their indices of statistical significance. In addition, I would avoid using acronyms in the abstract. If acronyms are deemed necessary, one might start using them in the introduction.

 In the method section, the authors state that “[a]n online quantitative survey, designed using psychometric principles and aligned with best practices for constructing an online assessment tool [32, 33], was shared with the participants through the university’s listserv.” It is unclear whether the researchers refer to both CPU and PWB questionnaires. If the survey contained both CPU and PWB, how was the survey presented to participants? A copy of both questionnaires may be added as an appendix to ensure clarity.

 What was the participation rate?

 The authors state on line 198 that “[t]ranslational validity, face validity, and content validity of the instrument was as-198 sessed before using the survey”. How did they measure translational validity?

The section entitled “data analysis” needs to be clarified. For instance, the sentence “[t]he correlation between CPU variables (CPSM-S and CPSM-O) and PWB was computed using ordinal logistic regression” is confusing. A correlation coefficient illustrates the relationship between two variables without taking into account others. An ordinal logistic regression examines the relative contribution of several variables to an outcome variable. I think the rationale of each analysis has to be clearly stated.

 In the result section, the outcome of each statistical analysis might be better organized if different paragraphs are labeled with the research question the analyses are intended to test.

The section labeled “Descriptive Statistics” contains inferential parametric statistics intended to describe group differences. Should the heading be changed? Furthermore, it is unclear why the authors use parametric inferential statistics in one section, and non-parametric ones in the other sections. If the authors assume that the variables of interest are measured at the ordinal level (instead of the interval level), then the median should be displayed along with a suitable measure of variability.

 Table 1 contains both means and medians. As noted earlier, if the authors assume that the variables of interest are measured at the ordinal level, then the median should be displayed along with a suitable measure of variability.

 In the discussion section, the connection between the literature reviewed in the introductory section and the researchers’ findings could be more thoroughly explained. Furthermore, the information regarding current results could be more efficiently organized and their implications explained more clearly. Overall, the discussion section leaves the reader to ask about the implications and applications of the current findings.

Please check the text for typos (e.g., “Procudures” on line 169). When a number starts a sentence, the number should be spelled out (see line 156). 

Author Response

(The authors gave the same response as above.)

Round 2

Reviewer 1 Report

The revised manuscript was much better. For the simple and clear principle, I have some suggestions.

1. In Line 229-230, the description of the item number and scorning method of the two scales in parentheses(CPSM-S (consisting of 8 items, scores ranging from 1-4) and CPSM-O (consisting of 4 items, scores ranging from 1-4)) have been told in the above paragrapth and better not present here. The author shoul check the full text to make sure it is consise.

2. For the ordinal logistic regression, please clearly point out what the categoris of PWB  which I point out in the last review report but the author did not directly answer it.

Author Response

Please refer to the rebuttal letter.
